# Assessment of Different Dimensions of Loneliness among Adults Living with Chronic Diseases

Dijana Babić [1,2,*], Snježana Benko Meštrović [3,4,5], Želimir Bertić [6,7], Milan Milošević [8] and Gordana Kamenečki [9]

1   Magdalena-Clinic for Cardiovascular Diseases, Ljudevita Gaja 2, 49217 Krapinske Toplice, Croatia
2   Department of Nursing, Catholic University of Croatia, Ilica 242, 10000 Zagreb, Croatia
3   Physiotherapy Department, Special Hospital for Pulmonary Diseases, Rockefellerova 3, 10000 Zagreb, Croatia; snjezanabenko@windowslive.com
4   Faculty of Health Studies, Viktora Cara Emina 5, 51000 Rijeka, Croatia
5   Department of Physiotherapy, University North, 104. Brigade 3, 42000 Varaždin, Croatia
6   Department of Nursing, Faculty of Health Studies, Viktora Cara Emina 5, 51000 Rijeka, Croatia; berticz@gmail.com
7   Department of Nursing, The Polytechnic of Ivanić Grad, Moslavačka 10, 10310 Ivanić-Grad, Croatia
8   Department for Occupational and Environmental Health, Andrija Stampar School of Public Health, School of Medicine, University of Zagreb, Rockefellerova 4, 10000 Zagreb, Croatia; milan.milosevic@snz.hr
9   Community Health Centre Krapina, Krapinsko-Zagorska County, Mirka Crkvenca 1, 49000 Krapina, Croatia; kameneckig@gmail.com
*   Correspondence: dijana.babic55@gmail.com

**Abstract: Background**: Loneliness has become a growing public health problem. Adult people who live with chronic health problems often experience more severe consequences of this condition. The purpose of this research is to determine the level of loneliness and differences in dimensions of loneliness in adults living with cardiovascular diseases (CVDs). **Methods**: The study was conducted in the Magdalena Clinic for Cardiovascular Diseases. The selected sample consisted of a group of patients admitted for short-term hospital treatment. The data were collected using the 11-item De Jong Gierveld Loneliness Scale (DJGLS). Differences in outcomes with respect to gender, educational level and marital, work and living status, as well as medical diagnoses of the participants, were compared using the Kruskal–Wallis test or the Mann–Whitney U test. Spearman's rho coefficient was used to analyze the correlation between the independent variables and the total score on different categories of loneliness. **Results**: A moderate level of loneliness (SD = 3.53; M = 3.0) was determined in almost half of the participants (N = 52; 49.1%). A statistically significant positive correlation was confirmed in the relationship between the emotional dimension of loneliness and the gender of the participants. The participants that had a lower education level showed a higher level of emotional loneliness, but also a higher overall level of loneliness. The age of the participants correlated negatively with the level of social loneliness (rho = −0.029). **Conclusions**: Loneliness is present among adults living with cardiovascular diseases in different dimensions and intensity. Although the connection between loneliness and health outcomes needs to be further investigated, the negative impact of this psychosocial problem on cardiovascular health cannot be ignored.

**Keywords:** loneliness; assessment; dimensions; adults; cardiovascular diseases





## 1. Introduction

In the past few years, public health authorities have been confronted with numerous challenges arising from demographic changes, economically driven migration, social deprivation and present post-pandemic life circumstances. The consequences of these are reflected in the significant increase in individuals living with multiple chronic health conditions like cardiovascular or malignant diseases, functional limitations, depression and other mental health problems, which all have a serious impact on overall well-being and quality of life. Among the health conditions listed, loneliness is gaining more scientific attention

considering its influence on day-to-day living. Loneliness can appear in all life stages, with a high frequency in adolescence, a decrease through middle adulthood, and finally a frequent reappearance in late adulthood (Beutel et al. 2017). According to evidence in the available literature, different degrees of loneliness are reported by 10.5% to 64.0% of adults older than 65 years (Beutel et al. 2017; Huang et al. 2021; Mullen et al. 2019; Ortiz-Ospina and Roser 2020; Chawla et al. 2021; World Health Organization 2021; Susanty et al. 2022). Theorists believe that a disproportion in a person's social needs and opportunities for social engagement and satisfaction, whether in a qualitative or quantitative sense, can precipitate the appearance of loneliness (Hawkley and Cacioppo 2010; Susanty et al. 2022).

Loneliness can be defined as a feeling of sadness and despair resulting from a lack of intimacy and contact with other people (Hammond and Pullen 2020). According to Peplau et al. (1979), loneliness is a perceived or experienced discrepancy between actual and desired social relationships (Steptoe et al. 2013; Polenick et al. 2021). Loneliness can be seen as a multidimensional and fluid construct that is comprised of two different components: the social and the emotional components (Hawkley et al. 2008; Ibáñez-del Valle et al. 2022; Manoli et al. 2022). Social loneliness represents a decrease in the quantity of social networks, while emotional loneliness refers to a reduction in the quality of social relations, i.e., a lack of closeness, understanding and trust between individuals (Dahlberg and McKee 2014; Guimarães Amorim et al. 2019). Studies have shown that the dimensions of loneliness change with age, but also due to specific life circumstances, whereby emotional loneliness increases during youth and in late adulthood, while social loneliness is stable in early and middle adulthood before suddenly decreasing in the final stages of life (Landmann and Rohmann 2022; Manoli et al. 2022). A third dimension of loneliness, called physical loneliness, has also recently been discussed in relevant works in the literature, and the research into this third dimension has been stimulated specifically by the COVID-19 pandemic and the imposed limiting of social contacts, that is, by the effects of these measures on psychosocial health and behavior (Landmann and Rohmann 2022).

The mechanisms of loneliness and its negative impact on an individual's well-being are widely discussed in the literature. There is an assumption that the long-term loss of reliable social ties can result in biases towards self-preservation and a heightened vigilance towards possible threats (Qualter et al. 2015; Holt-Lunstad et al. 2015). This might be a possible explanation for why individuals who feel lonely tend to evaluate current circumstances as more negative than they are, which exposes them to an increased level of psychological distress (Hawkley et al. 2010; Okruszek et al. 2020). The stated condition can further lead to an intentional separation from the social network and a loss of contact with others, consequently resulting with difficulties in physical and mental functioning (Holt-Lunstad et al. 2015).

Loneliness can significantly contribute to the occurrence of different health problems, especially in old age, including complex chronic conditions, functional decline and an increased risk of overall mortality (Holt-Lunstad et al. 2010; Hawkley and Cacioppo 2010; Tilvis et al. 2011; Theeke and Mallow 2013; Leigh-Hunt et al. 2017; O'Súilleabháin et al. 2019; Ortiz-Ospina and Roser 2020; Stokes et al. 2021; Ward et al. 2021). The physiological effects of loneliness develop over a relatively long time, most often through the mechanisms of negative health behavior, excessive reactivity to stress and ineffective physiological restoration processes (Hawkley and Cacioppo 2003; Leigh-Hunt et al. 2017).

Earlier studies have shown that the occurrence of loneliness significantly accelerates the process of physiological aging (Hawkley and Cacioppo 2007), and contributes to the occurrence of cardiovascular diseases such as ischemic heart disease, cerebrovascular incident and hypertension (Hawkley and Cacioppo 2007; Hawkley et al. 2010; Valtorta et al. 2016; Holt-Lunstad and Smith 2016; Xia and Li 2018; Donovan and Blazer 2020; Golaszewski et al. 2022) and numerous psychiatric conditions such as depression (Barg et al. 2006; Erzen and Çikrikci 2018; Achterbergh et al. 2020; Lee et al. 2021), alcoholism, sleep disorders, suicide, low self-esteem, Alzheimer's disease and chronic stress (Hawkley and Cacioppo 2010; Mushtaq et al. 2014; Beutel et al. 2017; Berg-Weger and Morley 2020; McLay et al.

2021). The association between loneliness and risk for cardiovascular health is explained by the interplay of different biological, psychological and sociological mechanisms (Valtorta et al. 2016; Xia and Li 2018; Hodgson et al. 2020; Bu et al. 2020). Studies have shown that a negative impact on the cardiovascular system most often occurs through impaired mechanisms of neuroendocrine regulation, the disruption of autonomic function and systolic pressure control, an elevated tendency towards an inflammatory response and higher levels of oxidative stress (Grippo et al. 2007; Cacioppo and Cacioppo 2015; Golaszewski et al. 2022). Furthermore, loneliness is often associated with the practice of negative health behaviors such as reduced physical activity, alcohol and nicotine consumption and inadequate nutrition, which present separate risk factors for cardiovascular diseases (Leigh-Hunt et al. 2017; Xia and Li 2018; Hodgson et al. 2020; Paul et al. 2021). The current data confirm that loneliness acts as a separate risk predictor for cardiovascular mortality, especially in women (Novak et al. 2020). The presence of an emotional dimension of loneliness significantly increases the risk for overall mortality, where the role of a mediator is attributed to the changes in the functional capacity of a person (O'Súilleabháin et al. 2019). Researchers have also confirmed that social isolation and loneliness contribute to an increased risk of heart attack and stroke, while the mortality related to these events increases in persons with a previously known history of CVD incidents (Hakulinen et al. 2018).

However, the existence of multiple chronic diseases significantly increases the risk of loneliness, especially in people who have experienced stressful life events and suffer from anxiety and depression (Stickley and Koyanagi 2018). The relationship between loneliness and physical health can be bidirectional (Ortiz-Ospina and Roser 2020). Previous research has shown that individuals who have experienced loneliness in life have a tendency to suffer from various diseases later on (Hawkley and Cacioppo 2010; Valtorta and Hanratty 2012; Ortiz-Ospina and Roser 2020), while on the other hand, individuals with impaired health can become lonely due to physical and other limitations, preventing them from achieving satisfying social contacts (Schrempft et al. 2019; Philip et al. 2020; Ortiz-Ospina and Roser 2020).

The identification and assessment of loneliness and social isolation often depend on understanding the nature of their origin, which represents the base for different shorter and longer versions of measuring instruments used in clinical and population studies (Manera et al. 2022). Among the most often used instruments are the 20-item Revised UCLA Loneliness Scale (Russell 1996), the 11-item De Jong Gierveld Loneliness scale (De Jong-Gierveld and Kamphuls 1985), the Berkman Social Network Index (Berkman and Breslow 1983) and the 10-item Lubben Social Network Scale (Lubben 1988). One of the key problems in the assessment of loneliness and social isolation is related to the heterogenicity of the items within the individual scales directed to the determination of social function deficits, without relying on clear theoretical origins that explain these social concepts (Manera et al. 2022), which makes the identification of results and possible differences difficult (Grenade and Boldy 2008). For the purposes of this study, we have selected the previously frequently used and psychometrically validated 11-item De Jong Gierveld's Loneliness Scale, which clearly identifies the emotional and social dimensions of this concept. Consequently, the aim of this study is to ascertain the overall level of loneliness in a selected sample, as well as the differences in the dimensions of loneliness in relation to the characteristics of the participants.

## 2. Materials and Methods

### 2.1. Participants and Settings

A cross-sectional observational study was conducted from March to August 2022 at the Magdalena Clinic for Cardiovascular Diseases in Krapinske Toplice. The selected sample was taken from middle-aged and old age patients who were admitted to the Clinic for short-term treatment due to diagnostic and interventional procedures. The inclusion criteria for participation in the study was the presence of at least one diagnosed CVD in the medical record, age $\geq$ 40 years, length of stay $\leq$ 5 days, expressed willingness to participate and an

ability to understand questions and communicate answers. After excluding incompletely filled out questionnaires, the study sample consisted of 106 randomly selected patients who met the inclusion criteria.

The total sample consisted of 70 men (66%) and 36 women (34%) (Table 1), with the average age being 68 years (SD = 8.51; CV = 13%). The level of education of more than half of the participants was high school (N = 54, 50.9%), while one third of them had a higher education level (N = 29, 27.4%). Almost two thirds of the participants were retired (N = 88, 83.0%), and only some of them still participated in a form of paid work (N = 18, 17.0%). The majority of the participants lived in a shared household, a majority of them with their spouses (N = 75, 70.8%). Among the clinical diagnoses present in the anamnesis, the most common were ischemic heart diseases (N = 53, 50.0%).

**Table 1.** Descriptive statistics of the socio-demographic variables and questionnaires (N = 106).

|  |  | **N** | **%** |
|---|---|---|---|
| Gender | Men | 70 | 66.0% |
|  | Women | 36 | 34.0% |
| Age | 40–60 | 13 | 12.2% |
|  | 61–70 | 55 | 50.0% |
|  | 71–83 | 38 | 35.8% |
| Education level | Elementary school or lower | 23 | 21.7% |
|  | High school degree | 54 | 50.9% |
|  | College or university degree | 29 | 27.4% |
| Retired | No | 18 | 17.0% |
|  | Yes | 88 | 83.0% |
| Living with partner | No | 31 | 29.2% |
|  | Yes | 75 | 70.8% |
| Financial income | Monthly salary | 15 | 14.2% |
|  | Pension | 91 | 85.8% |
| Diagnosis | Arterial hypertension | 14 | 13.2% |
|  | Ischemic heart disease | 53 | 50.0% |
|  | Other heart diseases | 34 | 32.1% |
|  | Peripheral arterial disease | 5 | 4.7% |
| De Jong Gierveld Scale | ≤2 (not lonely) | 47 | 44.3% |
|  | 3–8 (moderately lonely) | 52 | 49.1% |
|  | 9–10 (severely lonely) | 6 | 5.7% |
|  | 11 (very severely lonely) | 1 | 0.9% |

Source: medical records and structured questionnaire, 2022.

## 2.2. Variables and Measurements

The data were collected individually by educated registered nurses. The selected instruments were validated in Croatian and applied with the authors' approval. The assessment of loneliness was conducted using the 11-item De Jong Gierveld Loneliness Scale (De Jong-Gierveld and Kamphuls 1985). The scale consisted of a total of 11 statements, 6 of which indicated an emotional dimension of loneliness, while 5 of them referred to social loneliness. The scale uses response options on a 5-point Likert-type scale, where 0 = none of the time, 1 = rarely, 2 = some of the time, 3 = often and 4 = all of the time (De Jong-Gierveld et al. 2018). According to the method of scoring defined by the authors, positive answers (2 = some of the time, 3 = often and 4 = all of the time) to negative statements (items 2, 3, 5, 6, 9 and 10) provide an emotional loneliness score and negative answers (0 = none of the time, 1 = rarely, 2 = some of the time) to positive statements (items 1, 4, 7, 8 and 11) provide the social loneliness score. The maximum number of points that can be collected is 11, and a higher total of points indicates a higher severity of loneliness. This scale is a widely used tool for assessing loneliness in different age groups. Previous studies examining different samples, but mostly from older adults, have shown

an adequate level of reliability ($\alpha = 0.86 - 0.89$) and homogeneity (H = 0.47) of the 11-item DJGS, as well as the two subscales (emotional and social) (Grygiel et al. 2013; Iecovich 2013; Hosseini et al. 2021; Giraldo-Rodríguez et al. 2023). Data on the sociodemographic indicators and clinical diagnoses were collected using a structured questionnaire.

*2.3. Ethical Statement*

Our study has been approved by a suitably constituted Ethics Committee of the Magdalena Clinic for Cardiovascular Diseases under the number 195/INF-701/22 and conforms to the provisions of the Declaration of Helsinki in 1995 (as revised in Edinburgh 2000). Participants gave informed consent and patient anonymity was preserved.

*2.4. Statistical Analysis*

The results are presented in tables and figures. The categorical variables are shown as frequencies with the corresponding percentages. The differences in the calculated scores between different genders, education levels, work and living statuses, family status, financial resources, medical diagnoses and age groups were assessed with the $x^2$ test, Mann–Whitney U test or Kruskal–Wallis H test (in cases where more than two categories existed). All *p*-values below 0.05 were considered significant. The Cronbach alpha coefficient of internal consistency was calculated for the total score. Spearman's rho coefficient was used to analyze the correlation between the independent variables and the total score on different categories of loneliness. The statistical software IBM SPSS Statistics, version 25, was used in all statistical procedures.

**3. Results**

The assessment of loneliness showed a moderate level of loneliness in almost half of the participants (N = 52, 49.1%), while more severe forms of loneliness were rarely present and expressed by only a few participants (N = 7, 6.6%) (Table 1). Only one participant rated their loneliness as very severe (N = 1, 0.9%).

Table 2 shows the comparisons between the socio-demographic characteristics of the participants and the different dimensions of loneliness; that is, the total results reached on the DJGLS. As can be seen from the presented results, a statistically significant positive association was confirmed between the emotional dimension of loneliness and the gender of the participant (p1 = 0.005), where female participants expressed a higher level of emotional and overall loneliness (p3 = 0.020).

A very similar level of statistically significant positive correlation was confirmed between the level of education and emotional loneliness (p1 = 0.007), where participants with a lower level of education expressed a higher level of emotional as well as overall loneliness (p3 = 0.013). A difference in the feeling of loneliness based on work status, marital status and financial income was established, but this difference was not statistically relevant in any of the dimensions of loneliness.

Table 3 shows the results of the comparison between the dimensions of loneliness and the ages of the participants. A negative correlation between the age and the social dimension of loneliness has been noticed (rho = $-0.029$), but its significance is not statistically relevant ($p = 0.768$). The results also showed a negative correlation between living with a partner and emotional loneliness that was statistically significant (rho = $-0.247$; $p = 0.011$). A similar result was confirmed in relation to the overall level of loneliness (rho = $-0.216$; $p = 0.026$), through which it has been confirmed that individuals living in one-person households express a higher level of emotional and overall loneliness. The statistical association between the different dimensions of loneliness and overall loneliness has been confirmed with a high level of reliability ($p < 0.001$), which indicates the good mutual compliance of the variables within this measuring instrument.

**Table 2.** The differences between the different dimensions of loneliness and the sociodemographic characteristic of the sample: Kruskal–Wallis test/Mann–Whitney U test.

| | | N | Emotional Loneliness Score | | | Social Loneliness Score | | | Overall Loneliness Score | | | p1 | p2 | p3 |
|---|---|---|---|---|---|---|---|---|---|---|---|---|---|---|
| | | | Median | Percentile 25 | Percentile 75 | Median | Percentile 25 | Percentile 75 | Median | Percentile 25 | Percentile 75 | | | |
| Gender | Men | 70 | 1.00 | 1.00 | 3.00 | 1.00 | 0.00 | 2.00 | 2.00 | 1.00 | 4.00 | 0.005 | 0.326 | 0.020 |
| | Women | 36 | 3.00 | 1.00 | 4.00 | 1.00 | 0.00 | 4.00 | 4.50 | 1.50 | 6.50 | | | |
| Education level | Elementary school or lower | 23 | 3.00 | 2.00 | 5.00 | 2.00 | 0.00 | 4.00 | 5.00 | 3.00 | 8.00 | 0.007 | 0.095 | 0.013 |
| | High school degree | 54 | 1.00 | 1.00 | 3.00 | 0.50 | 0.00 | 2.00 | 2.00 | 1.00 | 5.00 | | | |
| | University degree | 29 | 2.00 | 1.00 | 2.00 | 1.00 | 0.00 | 2.00 | 3.00 | 1.00 | 4.00 | | | |
| Retired | No | 18 | 1.00 | 0.00 | 2.00 | 1.00 | 0.00 | 2.00 | 2.00 | 1.00 | 5.00 | 0.051 | 0.618 | 0.173 |
| | Yes | 88 | 2.00 | 1.00 | 3.00 | 1.00 | 0.00 | 2.00 | 3.00 | 1.00 | 6.00 | | | |
| Living with partner | No | 31 | 3.00 | 1.00 | 4.00 | 1.00 | 0.00 | 4.00 | 3.00 | 1.00 | 6.00 | 0.120 | 0.219 | 0.106 |
| | Yes | 75 | 2.00 | 1.00 | 3.00 | 1.00 | 0.00 | 2.00 | 3.00 | 1.00 | 5.00 | | | |
| Financial income | Monthly salary | 15 | 1.00 | 1.00 | 3.00 | 1.00 | 0.00 | 2.00 | 2.00 | 1.00 | 5.00 | 0.356 | 0.664 | 0.516 |
| | Pension | 91 | 2.00 | 1.00 | 3.00 | 1.00 | 0.00 | 2.00 | 3.00 | 1.00 | 6.00 | | | |
| Diagnosis | Arterial hypertension | 14 | 2.00 | 1.00 | 3.00 | 1.00 | 0.00 | 2.00 | 2.50 | 1.00 | 5.00 | 0.130 | 0.840 | 0.677 |
| | Ischemic heart disease | 53 | 2.00 | 1.00 | 3.00 | 1.00 | 0.00 | 2.00 | 3.00 | 1.00 | 6.00 | | | |
| | Other heart diseases | 34 | 2.00 | 1.00 | 3.00 | 1.00 | 0.00 | 2.00 | 3.00 | 1.00 | 5.00 | | | |
| | Peripheral arterial disease | 5 | 0.00 | 0.00 | 1.00 | 1.00 | 0.00 | 2.00 | 1.00 | 0.00 | 4.00 | | | |

Source: medical records and structured questionnaire, 2022.

**Table 3.** Correlation coefficients between the different dimensions of loneliness and age: Spearman's correlation coefficients.

| | | | Emotional Loneliness Score | Social Loneliness Score | Overall Loneliness Score |
|---|---|---|---|---|---|
| Spearman's rho | Emotional loneliness score | Correlation Coefficient | 1 | 0.482 | 0.874 |
| | | *p* | - | <0.001 | <0.001 |
| | | N | 106 | 106 | 106 |
| | Social loneliness score | Correlation Coefficient | 0.482 | 1 | 0.829 |
| | | *p* | <0.001 | - | <0.001 |
| | | N | 106 | 106 | 106 |
| | Overall loneliness score | Correlation Coefficient | 0.874 | 0.829 | 1 |
| | | *p* | <0.001 | <0.001 | - |
| | | N | 106 | 106 | 106 |
| | Age (years) | Correlation Coefficient | 0.151 | −0.029 | 0.087 |
| | | *p* | 0.121 | 0.768 | 0.376 |
| | | N | 106 | 106 | 106 |
| | Number of household members | Correlation Coefficient | −0.247 | −0.117 | −0.216 |
| | | *p* | 0.011 | 0.231 | 0.026 |
| | | N | 106 | 106 | 106 |

Source: medical records and questionnaire, 2022.

The reliability analysis of the statements about loneliness made using the Cronbach alpha model has shown a good degree of reliability ($\alpha = 0.837$), which indicates the proper alignment of individual statements and answers within this scale.

## 4. Discussion

Loneliness represents a complex set of feelings which have a significant impact on different dimensions of an individual's health and overall well-being. We determined the presence of a moderate level of loneliness in the majority of respondents, but also certain differences in the dimensions of loneliness in relation to the individual characteristics of the participants.

Earlier studies have confirmed the connection between loneliness and the occurrence of serious health problems such as depression, diabetes, autoimmune diseases, obesity, reduced functional capacity, sleep disorders, problems with mental health and cognition and incidents related to cardiovascular health (Hawkley and Cacioppo 2003; Cacioppo et al. 2010; Mushtaq et al. 2014; Valtorta et al. 2016; Luo et al. 2012; Golaszewski et al. 2022). The relationship between cardiovascular health and feelings of loneliness is still a subject of scientific research. The connection between these two states is based on the fact that exposure to repeated and chronic stress encourages the development of negative health behaviors (smoking, lack of physical activity), activates biological mechanisms (inflammatory response, oxidative stress, damage to reparative processes) and increases the influence of psychological factors (depressed mood) which may result from the development of cardiovascular symptoms and disorders (Hawkley and Cacioppo 2003; Xia and Li 2018; Paul et al. 2021; Li and Xia 2020). A study conducted by Patterson and Veenstra (2010) showed that reduced physical activity and the appearance of a depressed mood act as

mediators in the relationship between loneliness and the occurrence of chronic health problems.

The results of our research showed a relatively moderate level of loneliness among participants who already had one or more chronic cardiovascular disease. Possible reasons that would explain this result can be found in the sociodemographic characteristics of the examined sample, which show that almost two thirds of the participants had a spouse and lived in households with family members. Although it has been proven that the existence of one or more cardiovascular diseases has a negative impact on functional capacity (Arena et al. 2014; Forman et al. 2017; Fuentes-Abolafio et al. 2020) and participation in social activities (Piepoli et al. 2019), most often through the mechanisms of a reduction in physical strength, lack of self-confidence, difficulties with coping and the need for the daily intake of numerous medications (Leeming et al. 2014), the availability of informal social support (i.e., from the spouse, relatives or neighbors) can reduce the negative effects of the aforementioned factors. A study by Wenn et al. (2022) confirmed that the availability of social support significantly affected the quality of life of patients with heart diseases, encourages compliance with therapeutic measures and the avoidance of negative health behaviors.

Previously conducted studies show that feelings of loneliness increases with advanced age, where risk factors such as the loss of family ties, partnership or friendships, lower mobility, lack of income and health conditions are stated (Fakoya et al. 2020; Takagi et al. 2020; Czaja et al. 2021). The results of our research showed a negative correlation between age and the appearance of social dimensions of loneliness. The research conducted by Singh and Misra (2009) showed that the possession of intellectual and social resources acquired during employment as well as previous life experiences have an important protective role in the prevention of loneliness in older age. The study also proved that these protective mechanisms are more present among older men who were previously socially active, because they allow them to successfully maintain the habits of established contacts and new social connections (Singh and Misra 2009). Since the sample in our research consisted mostly of male individuals, the results can partially be explained with the respective findings. At the same time, it is also necessary to contextualize the role of the spouse or partner, a proven protective mechanism in the prevention of loneliness.

The emotional but also overall level of loneliness was significantly pronounced in female participants. A possible explanation for this result lies in the fact that female individuals experience the emotional dimension of social relationships more intensely, since they can be more inclined to clearly show their emotions (Deng et al. 2016; Fischer et al. 2018), and often appreciate emotional intimacy more than men (Birditt and Fingerman 2003). Older men are, on the other hand, often constrained by the roles and expectations that society places on them, which consequently creates problems with articulating and expressing feelings of loneliness (Ratcliffe et al. 2024).

Loneliness significantly affects the self-perception of physical and mental health, especially for women (Boehlen et al. 2022). Earlier studies have shown that loneliness significantly affects the subjective and objective understanding of health, but also health outcomes in people suffering from chronic diseases. Although loneliness has been proven to contribute to a wide range of different health risks, including mortality of any causes and incident events in persons living with CVD (Paul et al. 2021), this phenomenon is particularly pronounced in women. A study by Golaszewski et al. (2022) found that older women with social isolation and loneliness had a 13 to 27% higher risk of a CVD incident than those with lower social risk factors. A study conducted by Novak et al. (2020) confirmed similar findings, stating that loneliness is an independent predictor of cardiovascular mortality in older women, but not in men.

Therefore, the early detection of individuals at risk is very important, especially women living with CVD, with the aim of preventing negative health outcomes and alleviating the overall effect on the health status. The results of our study also showed a significant difference in the expression of loneliness with regard to the education level of the partic-

ipants. We confirmed the previously known fact that people with a higher educational profile expressed a lower level of loneliness. Similar findings were confirmed by other studies (Hawkley et al. 2008; Cohen-Mansfield et al. 2016; Nguyen et al. 2020; Fierloos et al. 2021), whose authors see a possible explanation of these results through a positive effect of education on the exposure to chronic stress, the opportunity for the construction of a larger social network, a better quality of social and interpersonal relationships and more favorable socioeconomic circumstances. However, Neto (2014) claims that the level of education cannot be seen as an independent predictor of loneliness, because factors such as marital status and age often act as mediators that reduce the negative effect of this feeling. Unexpectedly, contradictory results were obtained in the study by Nicolaisen and Thorsen (2012), where it was visible that an increase in age and the level of education of the participants had a positive correlation with the levels of loneliness. A higher level of education enables a higher level of expertise and mastery of certain skills, better health and quality of life, as well as greater chances of living with a partner in old age, as explained by the authors of the study. Higher levels of education are associated with higher levels of health literacy and a better comprehension of health information and orientation in the health system (Jansen et al. 2018). This fact is especially important for patients with chronic cardiovascular diseases, because a better understanding of health information often contributes to better adherence to the therapeutic plan, allows easier cooperation between patients and health professionals and ensures better health outcomes. According to a study by Vasan et al. (2023), low health literacy is associated with an increased sense of loneliness, where the negative impact is manifested through the adoption of negative health behaviors, e.g., greater perceived stress, even when controlling other psychosocial risk factors.

A detailed analysis of the statements representing the emotional dimension of loneliness showed a possible deficit of solid, meaningful social ties that are created throughout life. Tiilikainen and Seppänen (2017) report that the causes of emotional loneliness in old age are found in the loss or unfulfillment of deeper emotional connections, such as the lack of a life partner, an absence of meaningful friendships and complex parenting or problematic childhood experiences. The study conducted by Kim et al. (2022) also confirmed a connection between loneliness in later life and the characteristics of previous social relationships. The authors explained the importance of changes in marital and work status and their contribution to the occurrence of this phenomenon through the effects of the reduction and reorganization of an individual's social network. However, satisfaction with social connections can have a mediating role because individuals who present a higher level of satisfaction with their own network generally report the absence of or lower levels of loneliness (Kemperman et al. 2019).

Among the statements that indicate the social aspect of loneliness, those often chosen were ones that indicate a lack of close friends and trusted persons. Earlier studies often found an association between loneliness, a lower quality of social relationships and a deficit of social support with high levels of depression (Schwarzbach et al. 2014; Donovan et al. 2016; Taylor et al. 2018; Donovan and Blazer 2020). These data may indicate that individuals who are depressed or feel depressed due to their present health condition and imposed circumstances can experience their social relationships and sources of social support less adequately or deficiently, with a reduced sense of closeness (Donovan and Blazer 2020). The sample of participants that were included in this research was mostly made up of adult individuals who were staying in a specific environment and were going through the diagnostics process and treatment for a chronic illness. These events could have affected their self-perception and their experience of themselves, their health and their relationships with others, especially since their current circumstances could cause concern, fear and feelings of uncertainty and powerlessness, thus additionally enhancing the feeling of loneliness.

A study conducted by Dahlberg (2007) emphasizes that the concept of loneliness always includes feelings of rejection or abandonment, which can be voluntary or imposed, and which are determined by the absence of an important "other" person and connections

with significant others. Likewise, loneliness can be present when a person believes that people with whom they feel a strong interpersonal connection are not available for social interaction and support, as well as in circumstances that cause a low value of relationships with others, such as the loss of a spouse, the breakup of a relationship or relocation (Leary 2015). A theoretical model described in the relevant literature called the "Acceptance-Rejection Theory" (Rohner 2021) explains the influence of the subjective perception of the quality of interpersonal relationships during the lifetime on psychological well-being (Senese et al. 2021). According to this model, individuals who cannot satisfy their need for love and acceptance by others develop a psychological state that is characterized by a greater likelihood of depression, substance abuse, fear of intimacy and loneliness (Senese et al. 2021). Based on this assumption, it can be said that the feelings of rejection or disapproval, which were also detected in our research, can be a predictor or indicator of the early phases of loneliness.

We need to address the limitations of this study. In the first instance, we need to point out the relative homogeneity and size of the selected sample. The sample consisted of individuals admitted for a short-time hospital treatment, which was always planned according to the severity of illness and complexity of current health status. This assumes that these were not people whose physical or mental health was seriously compromised at the time of research, which partially explains the obtained results. Under these conditions, the limited sample size did not give a result that is representative for the total population of CVD patients. Also, we noticed that it took different amounts of time to fill out the questionnaire, especially for older individuals, mostly because they often needed clarifications of certain concepts and assistance with completing the questionnaire. Therefore, it was necessary to consider the use of a shortened version of DJGLS or a similar instrument in research among an older population.

Although this research has limitations, the findings present an amendment to previous knowledge regarding the occurrence of loneliness in adult or older adult individuals. The presence of chronic cardiovascular disease provides a negative effect on individuals' physical or mental health, and introduces a person to new circumstances which are often characterized by anxiety, fear and feelings of insecurity, mostly related to uncertain treatment outcomes. The absence of support in such situations can precipitate feelings of loneliness, isolation and abandonment from others. Higher levels of importance should be given to the development of preventive programs aimed towards emotional support systems for women who live with chronic diseases, especially those living without a partner or family. Future studies also need to direct a focus to people with a lower educational status and limited financial and social resources who did not have the opportunity to build a network of social and emotional support throughout their lives.

## 5. Conclusions

The findings of the conducted study detected the presence of moderate levels of loneliness among a selected sample of individuals who live with chronic cardiovascular disease. However, certain differences in dimensions of loneliness and personal characteristics of the respondents were observed. Special attention should be given to female individuals and those with advanced age who live with chronic health problems and limited financial and social resources, which makes them particularly susceptible to the occurrence of loneliness. Although the impact of loneliness on health status and health outcomes has not been investigated, it should not be ignored. We believe that these findings can improve our understanding of loneliness as a public health and psychosocial problem which requires a systematic and comprehensive approach. Further efforts should be made in the field of research into different dimensions of loneliness and the levels of influence they have on the health and overall well-being of individuals living with chronic cardiovascular diseases.

**Author Contributions:** Conceptualization, D.B. and G.K.; methodology, D.B.; software, M.M.; validation, M.M. and Ž.B.; formal analysis, M.M.; investigation, D.B., G.K. and S.B.M.; resources, D.B., Ž.B. and S.B.M.; data curation, D.B. and G.K.; writing—original draft preparation, D.B.; writing—review

and editing, Ž.B. and S.B.M.; visualization, D.B.; supervision, G.K. and M.M.; project administration, D.B.; funding acquisition, D.B. and S.B.M. All authors have read and agreed to the published version of the manuscript.

**Funding:** This research received no external funding.

**Institutional Review Board Statement:** The study was conducted in accordance with the Declaration of Helsinki, and approved by the Institutional Review Board (or Ethics Committee) of the Magdalena Clinic for Cardiovascular Diseases (protocol code 195/INF-701/22).

**Informed Consent Statement:** Informed consent was obtained from all subjects involved in the study.

**Data Availability Statement:** The data presented in this study are available on request from the corresponding author. The data are not publicly available due to due to the protection of the respondent's right to confidentiality.

**Conflicts of Interest:** The authors declare no conflict of interest.

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
