# Peer review of "Assessment of Different Dimensions of Loneliness among Adults Living with Chronic Diseases"

_socsci, doi:10.3390/socsci13040202_

Round 1
Reviewer 1 Report
Comments and Suggestions for Authors
Dear Authors,
I'm sorry but your work have originality but important and insurmontable methodologycal defects:
- in The Title is indicated generic Chronic Diseases bnut in the text appears only Cardiovascular sample;
- The Buckground is vague and haven't the structure and the report can support the idea and the objective (I suggest a specific introduction to Loneliness and Cardiovascular Disease and not a generic cronic disease;
- The method of selection af sample not declared (missing the specific inclusion and exclusion criteria);
- There is a stratification of Cardiovascular Disease but missing for single choort the level of severity of the illness (possibile bias?!...);
- Missing the limitation of the study (popolation, choort, stratification...);
- The discussion always generic and not specific for Cardiovascular Disease;
- The conclusion are not in line with objective of study.
Best Regards.
Author Response
Dear Mrs./Mr.,
We are grateful for all your comments and suggestions. We hope that we have adequately answered to your requests and that this version of the article will be in order with the instructions.
With kind regards,
Authors

Reviewer 2 Report
Comments and Suggestions for Authors
The paper addresses the issue of loneliness in old age. The topic is topical but the paper leaves some perplexity.
First of all, from a methodological point of view, the description of the instrument used to assess loneliness, even considering the theoretical premise about the existence of different instruments differing in the constructs used, should be expanded to include psychometric indications guaranteeing the goodness of the instrument, threshold values or reference data by age and theoretical justifications underlying the choice of the instrument in issue. Secondly, the sample is unbalanced not only by gender but also by age and is first described according to the mean and standard deviation to which is added the Coefficient of Variation whose use in the context is doubtful since not two distributions but only one compared. In Table 1, the subjects are divided into 3 age groups with a lower age group between 40 and 60; in Table 2, we discover that the maximum age is 83 but the minimum is 28! Nowhere in the paper is the decision to eliminate outliers from the analyses explicitly mentioned. Moreover, already in the abstract (lines 17 and 18) the existence of a negative correlation between age and social loneliness is emphasised, a concept that is taken up again in lines 163-165 where, however, it is admitted that the correlation is not statistically significant. The discussion is very articulate and accompanied by bibliographical references, but it is based on data that, as presented and analysed in this version, appear inconsistent and non-generalisable: perhaps it would be appropriate to limit the analysis to a subgroup that is more homogeneous in terms of age and gender and verify the consistency of the results.
Comments on the Quality of English LanguageUse a corrector for the English language to avoid trivial typing errors (i.e. Ishaemic instead of Ischaemic)
Author Response
We are grateful for all your comments and suggestions. We hope that we have adequately answered to your requests and that this version of the article will be in order with the instructions.
With kind regards,
Authors

Reviewer 3 Report
Comments and Suggestions for Authors
This manuscript aims to assess to ascertain the overall level of loneliness among adults living with chronic diseases. While the study has the potential to positively motivate future research, the findings are not necessarily novel, nor does the research contribute many novel considerations and examinations of such results, especially clinical implications and discussion for how to potentially improve quality of life in this population. Furthermore, the overall structure, organization and grammar of the writing in the manuscript detracts significantly from the overall meaning.
Abstract
Authors should add information about: place of study, study group and remove the P values.
Introduction
The introduction has been described extensively.
Material and Methods
- Inclusion and exclusion criteria need to be included, and study design needs to be specified
- Please describe the selection of the study group - on what basis the size was calculated
- Please describe the materials and methods in detail, listing in turn: 2.1. Participants and Settings, 2.2. Variables and Measurements, 2.3. Ethical Statement, 2.4. Statistical Analysis
Results
- Descriptive study should be in material and methods
- Maximum, Minimum, percentile do not contribute anything
- no explanations under the tables
Discussion
- write limitations of study
Author Response

(The authors gave the same response as above.)

Round 2
Reviewer 1 Report
Comments and Suggestions for Authors
Dear Authors,
Answer 1, check;
Answer 2, sorry for possible misununderstanding but with Background I considered the Introdution of the text (see Answer 6);
Answer 3, check;
Answer 4, sorry for the unclear but you answered with Answer 5;
Answer 5, check;
Answer 6, check;
Answer 7, check.
Best regards.
Author Response
Please, see the attachment.
Best regards

Reviewer 2 Report
Comments and Suggestions for Authors
The authors positively answer to the done observations. The paper can be now published.
